# Masked Autoencoders As Spatiotemporal Learners

**Christoph Feichtenhofer**[*]    **Haoqi Fan**[*]    **Yanghao Li**    **Kaiming He**
Meta AI, FAIR
https://github.com/facebookresearch/mae_st

## Abstract

This paper studies a conceptually simple extension of Masked Autoencoders (MAE) [31] to spatiotemporal representation learning from videos. We randomly mask out spacetime patches in videos and learn an autoencoder to reconstruct them in pixels. Interestingly, we show that our MAE method can learn strong representations with *almost no inductive bias* on spacetime (only except for patch and positional embeddings), and spacetime-*agnostic* random masking performs the best. We observe that the optimal masking ratio is as high as 90% (*vs*. 75% on images [31]), supporting the hypothesis that this ratio is related to information redundancy of the data. A high masking ratio leads to a large speedup, *e.g*., $> 4\times$ in wall-clock time or even more. We report competitive results on several challenging video datasets using vanilla Vision Transformers [18]. We observe that MAE can outperform supervised pre-training by large margins. We further report encouraging results of training on real-world, uncurated Instagram data. Our study suggests that the general framework of masked autoencoding (BERT [15], MAE [31], *etc*.) can be a unified methodology for representation learning with minimal domain knowledge.

## 1 Introduction

The deep learning community is experiencing a trend of unifying methodologies for solving problems in different areas, such as language, vision, speech, and more. For architectures, Transformers [67] have been successfully introduced into computer vision [18] and established as a general building block in both language and vision. For self-supervised representation learning, the *denoising/masked autoencoding* methodology [68] in BERT [15] has been shown effective on learning visual representations from images [31]. Towards unifying methodologies, less domain knowledge ("fewer inductive biases" [18]) is introduced for a specific problem, which urges the models to learn useful knowledge almost purely from data.

Following this philosophy, we study extending Masked Autoencoders (MAE) [31] to the problem of spatiotemporal representation learning. Our method is simple: we randomly mask out spacetime patches in videos and learn an autoencoder to reconstruct them (Fig. 1). Our method has *minimal* domain knowledge: the only spacetime-specific inductive bias is on embedding the patches and their positions; all other components are *agnostic* to the spacetime nature of the problem. In particular, our encoder and decoder are both vanilla Vision Transformers [18] with no factorization or hierarchy, and our random mask sampling is agnostic to the spacetime structures. Our method predicts pixel values and uses no extra problem-specific tokenizer. In a nutshell, our method is simply MAE applied to the set of spacetime patches. Despite minimal inductive biases, our method achieves strong empirical results, suggesting that useful knowledge can be *learned from data*.

It is hypothesized in [31] that the masking ratio (*i.e*., percentage of removed tokens) in masked autoencoding methods is related to the information redundancy of the problems. For example, natural images are more information-redundant than languages and thus the optimal masking ratio is higher (*e.g*., than BERT [15]). Our observations on video data support this hypothesis. We find that the optimal masking ratio of MAE is 90% for videos (Fig. 2), higher than the masking ratio of 75% for its image counterpart [31]. This can be understood as a consequence of natural video being correlated.

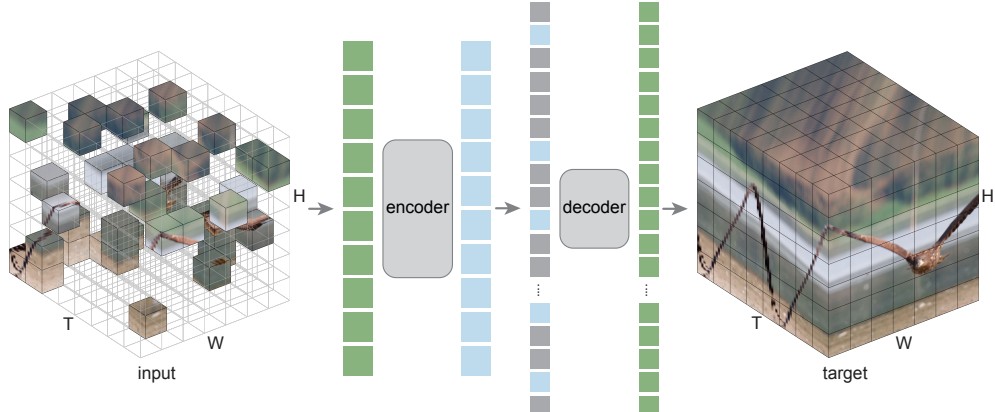

Figure 1: **Masked Autoencoders as spatiotemporal learners**. We mask a large subset (*e.g.*, 90%) of random patches in spacetime. An encoder operates on the set of visible patches. A small decoder then processes the full set of encoded patches and mask tokens to reconstruct the input. Except for patch and positional embeddings, *neither the encoder, the decoder, nor the masking strategy, has any spatiotemporal inductive bias*.

To the extreme, if a video has $T$ identical static frames, randomly sampling $1/T$ of all spacetime patches would reveal most of the static frame. Because slow motion is more likely than fast motion in natural videos, the masking ratio can be very high as we observe empirically.

The higher masking ratio leads to a more efficient solution in practice. Following the MAE in [31] that applies the encoder only on visible tokens, a masking ratio of 90% reduces the encoder time and memory complexity to $<1/10$. Put together with a small decoder [31], the MAE pre-training can achieve a theoretically $7.7\times$ reduction in computation *vs.* encoding all tokens. In fact, the computation reduction is so large that the data loading time becomes a new bottleneck; even so, we record a $4.1\times$ wall-clock speedup. Such a significant speedup is of great importance for video research that is large-scale and time-consuming.

We report strong results on a variety of video recognition datasets. Our MAE pre-training greatly improves generalization performance: on Kinetics-400 [35], it increases the accuracy of ViT-Large [18] by absolute 13% *vs.* training from scratch, while it takes *less* wall-clock training time overall (pre-training plus fine-tuning). Our MAE pre-training can outperform its supervised pre-training counterpart by big margins. Using vanilla ViT [18], our method achieves competitive results with previous state-of-the-art methods that incorporate more domain knowledge. We also report encouraging results using MAE pre-trained on 1 million random, *uncurated* Instagram videos. These results suggest that self-supervised learning on videos can be tackled in a way similar to its counterparts on language [15] and images [31], under a unified framework.

## 2 Related Work

**Denoising autoencoders.** Denoising autoencoders (DAE) [68, 69] present a general methodology for learning representations by reconstructing clean signals from corrupted inputs. Masking as a type of noise dates back to at least a decade ago [69]. One of its most successful developments is BERT [15], which is conceptually masked autoencoding on language tokens.

Denoising/masked autoencoding methods for computer vision have been making continuous progress [50, 9, 18, 31]. A series of recent methods are based on Transformer architectures [67] and are towards a unified solution between vision and language. iGPT [9] pioneers this direction by training Transformers on pixels as tokens. The ViT paper [18] makes a revolutionary step forward by using patches as tokens. It not only establishes strong Transformer architectures for vision tasks, but also explores masked prediction with patches. MAE [31] returns to the basics of the autoencoding concept [68] and draws attention to the decoding aspect. The presence of a meaningful decoder provides more flexibility, *e.g.*, enabling the encoder to operate only on visible patches and leading to a more efficient solution. It empirically shows that a high masking ratio is essential for image tasks [31]. Our study follows this line of research.

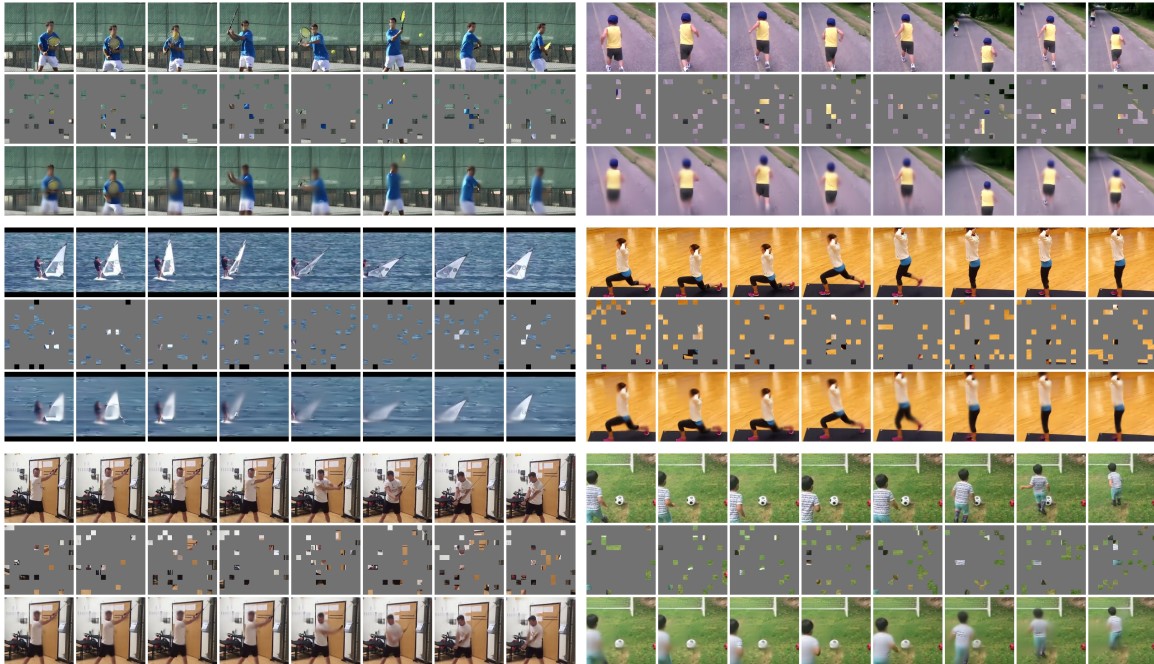

Figure 2: Visualizations on the Kinetics-400 [35] validation set (masking ratio **90%**). We show the original video (top), masked video (middle), and MAE output (bottom) for each sample. This model reconstructs the original pixels. The video size is $16{\times}224{\times}224$ and the spacetime patch size is $2{\times}16{\times}16$ (the temporal patch size of 2 is not visualized here). Each sample has $8{\times}14{\times}14{=}1568$ tokens with 156 being visible. For better visualizations, the known patches in the output are from the original input. Fig. 7 shows more examples.

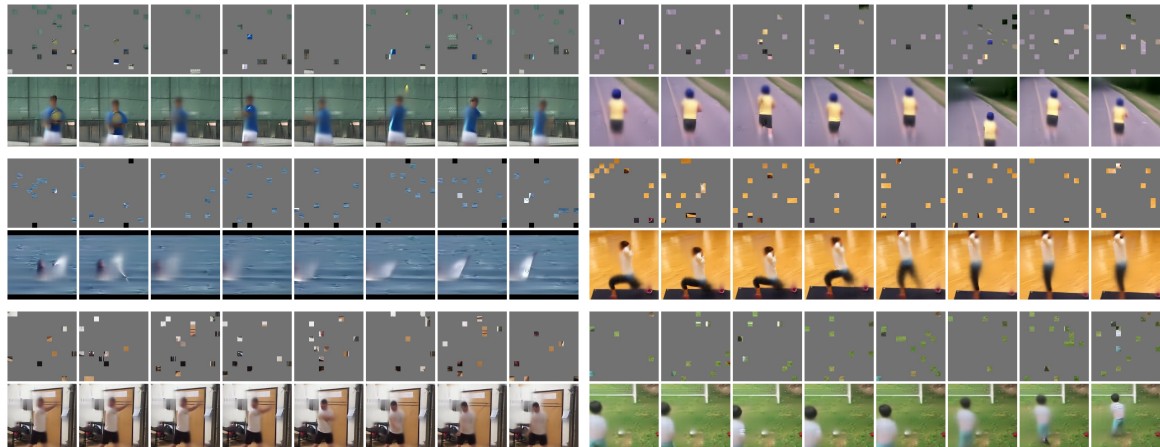

Figure 3: Visualizations of the same pre-trained model in Fig. 2 but with a masking ratio of **95%**.

Instead of predicting pixels [9, 18, 31, 80], another line of research focuses on the tokenization of the prediction targets [3, 17, 77]. BEiT [3] proposes to use pre-trained dVAE [47, 55] as the reconstruction target. The dVAE tokenizer can be improved by perceptual or adversarial losses [17]. MaskFeat [77] shows that HoG [13] as prediction targets performs strongly.

**Self-supervised learning on videos.** The presence of the temporal dimension is a focus of self-supervised learning on video data. Related topics include temporal coherence ('slowness') [79, 25], future prediction [61, 72, 70, 45, 44, 71, 16], object motion [1, 75, 49, 76], temporal ordering [46, 23, 38, 78, 81], spatiotemporal contrast [58, 62, 30, 22, 51, 56], *etc.*

Our method also relies on the temporal coherence of videos, but it approaches this goal implicitly. In fact, as our method is largely agnostic to spacetime, the main opportunity for it to make use of the temporal coherence is a *higher* masking ratio (*e.g.*, 90%), which assumes that videos are more information-redundant than images.

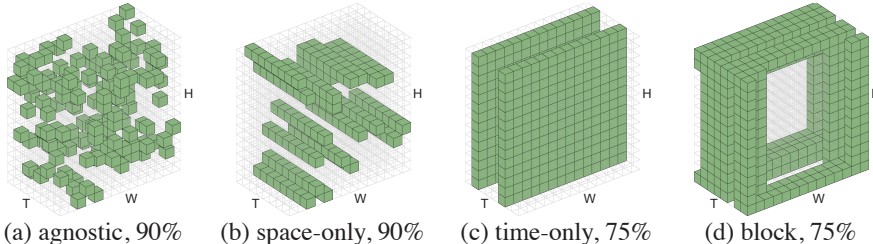

| (a) agnostic, 90% | (b) space-only, 90% | (c) time-only, 75% | (d) block, 75% |

Figure 4: **Mask sampling**. **(a)**: Random sampling that is spacetime-*agnostic*. **(b)**: Space-only random sampling, broadcasted to all time steps ("tube" masking [77]). **(c)**: Time-only random sampling, broadcasted to all spatial locations ("frame" masking [77]). **(d)**: Block-wise sampling [3] in spacetime, removing large regions ("cube" masking [77]). In this illustration, $T{\times}H{\times}W$ is $8{\times}14{\times}14$; green tokens are kept and others are masked out.

There has been growing interest in masking-based methods for self-supervised learning on videos. Previous works focus on tokenizing the prediction targets for the use of videos [65, 73, 77]. Our autoencoding method operates on pixels, which is simpler and requires no extra data or domain knowledge on the tokenizer. Importantly, our method greatly improves the *efficiency* of learning. The practical speedup is of central importance for video-related research, which is in general larger-scale and more time-consuming.

Our work is done independently and concurrently with [66] on a related method.

## 3   Method

Our method is a simple extension of MAE [31] to spacetime data (Fig. 1). Our goal is to develop the method under a general and unified framework, with as little domain knowledge as possible.

**Patch embedding.** Following the original ViT [18], given a video clip, we divide it into a regular grid of non-overlapping patches in spacetime [4, 2, 19, 77]. The patches are flattened and embedded by linear projection [18]. Positional embeddings [67] are added to the embedded patches. The patch and positional embedding process is the only process that is spacetime-aware.

**Masking.** We sample random patches without replacement from the set of embedded patches. This random sampling is *agnostic* to the spacetime structure (Fig. 4 (a)). This structure-agnostic sampling strategy is analogous to that of BERT in 1D [15] and MAE in 2D [31].

It is hypothesized in [31] that the optimal masking ratio is related to the information redundancy of the data. With unstructured random masking, BERT [15] uses a masking ratio of 15% for language and MAE [31] uses a ratio of 75% for images, suggesting that images are more information-redundant than language. Our empirical results on videos support this hypothesis. The optimal masking ratio we observe is 90%. This is in line with the common assumption that natural videos are more information-redundant than images because of temporal coherence. Fig. 2 and 3 present our MAE reconstruction results on unseen validation data with a masking ratio of 90% and 95%.

The spacetime-agnostic sampling can be more effective than structure-aware sampling strategies, *e.g.*, *space-only*, *time-only*, or *block-wise* sampling (Fig. 4 (b-d)). As neighboring patches in space or in time (Fig. 4(b, c)) are coherent, with a very high masking ratio, space-only or time-only sampling may retain less information and yield an overly difficult pre-training task. For example, time-only sampling from 8 frames with a masking ratio of 87.5% means keeping only a single frame, which presents an overly challenging task of predicting the future and past given only one frame. We observe that optimal masking ratios for structure-aware sampling are in general lower. In contrast, the spacetime-agnostic sampling better utilizes the limited number of visible patches and thus allows to use a higher masking ratio.

**Autoencoding.** Our encoder is a vanilla ViT [18] applied only on the visible set of embedded patches, following [31]. This design greatly reduces time and memory complexity and leads to a more practical solution. A masking ratio of 90% reduces the encoder complexity to $<1/10$ (noting that self-attention is quadratically-complex w.r.t. the token set size).

Our decoder is another vanilla ViT on the union of the encoded patch set and a set of mask tokens [31]. Decoder-specific positional embeddings are added to this set [31]. The decoder is designed to be smaller than the encoder [31]. Although the decoder processes the full set, its complexity is smaller than the encoder (*e.g.*, ~1/20 per token). In our default setting, the overall autoencoder has a complexity reduction of $7.7\times$ *vs.* full encoding (more discussions are in Sec. 5.1 and Table 1).

The decoder predicts the patches in the *pixel* space. In principle we can simply predict a full spacetime patch (*e.g.*, $t\times16\times16$); in practice, we find it sufficient to predict a single time slice of the patch ($16\times16$), which keeps the prediction layer's size manageable. We predict the original pixels or their per-patch normalized values [31] (compared in Table 2b). The training loss function is the mean squared error (MSE) between the prediction and its target, averaged over unknown patches [15].

The encoder and decoder are agnostic to the spacetime structure of the problem. There is *no* hierarchy or spacetime factorization, in contrast to the leading architectures [4, 2, 19]. Our method relies on the global self-attention to learn useful knowledge from data, following the spirit of [18].

# 4    Implementation

**Data pre-processing.** For MAE pre-training, our default input size is 16 frames each with $224\times224$ pixels (*i.e.*, $16\times224\times224$). The 16 frames are sampled from the raw video with a temporal stride of 4 (*i.e.*, $16\times4$ sampling in the literature [21]), and the starting frame is randomly sampled. In the spatial domain, we perform random resized cropping [63] with a scale range of $[0.5, 1]$, and random horizontal flipping. We do *not* apply other data augmentations unless noted.

Our MAE pre-training is so fast in computation that data loading becomes a new bottleneck that dominates running time in our setup. We adopt *repeated sampling* [33][1] to alleviate this problem. Each time a raw video is loaded and decompressed, we take multiple (4 by default) samples from it. This reduces the data loading and decompressing time per sample. We note that repeated sampling does *not* change the number of samples seen; it only influences the *orders* of the samples seen during training. We always count epochs as "effective epochs", *i.e.*, how many times each raw video is sampled throughout training.

**Architecture.** Our encoder and decoder are the *vanilla* ViT architectures [18]. We use a temporal patch size of 2 [2, 19, 77] and a spatial patch size of $16\times16$ [18], denoted as $2\times16\times16$. We use the same patch size for ViT-B/L/H [18] for simplicity. For a $16\times224\times224$ input, this patch size produces $8\times14\times14$ tokens.

We adopt separable positional embeddings for the encoder. We have two positional embeddings, one for space and the other for time. The spacetime positional embeddings are the sum of them. This separable implementation prevents the size of positional embeddings growing too large in 3D. We use learnable positional embeddings; the sin-cos variant [67] works similarly.

**Settings.** Our MAE pre-training configuration mostly follows [31]. We use the AdamW optimizer [43] with a batch size of 512. We evaluate the pre-training quality by end-to-end fine-tuning. The choice of evaluating by fine-tuning (instead of linear probing) follows [3, 31]. Our inference process follows the common practice of multi-view testing [74, 21]: it takes $K$ temporal clips (by default $K{=}7$ on Kinetics) to cover the video length, and for each clip it takes 3 spatial views to cover the longer spatial axis (denoted as $K\times3$). The final prediction is the average of all views. The implementation details and hyper-parameters are in the appendix.

# 5    Experiments

In Sec. 5.1 and Sec. 5.2 we perform ablation experiments on Kinetics-400 (K400) [35]. We do MAE self-supervised pre-training and then fine-tune the encoder with supervision for evaluation. We report top-1 classification accuracy (%) on the K400 validation set. In Sec. 5.3 we study more pre-training datasets and downstream tasks.

---

[1]In our use case, repeated sampling involves data augmentation and mask sampling.

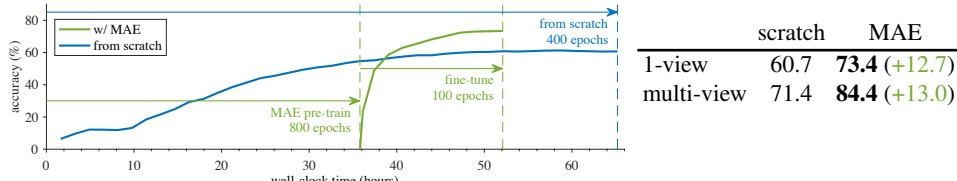

Figure 5: MAE pre-training plus fine-tuning is *much more accurate* and *faster* than training from scratch. Here the x-axis is the wall-clock training time (128 A100 GPUs), and the y-axis is the 1-view accuracy on Kinetics-400 validation. The table shows the final accuracy. The model is ViT-L.

| MAE w/ | acc. | FLOPs | compute | load+compute |
|---|---|---|---|---|
| encoder w/ [M] | 84.3 | 627.5 G | 141.1 hr | 147.5 hr |
| encoder w/o [M] | 84.4 | 81.0 G | 24.5 hr | 35.8 hr |
| gain | | 7.7× | 5.8× | 4.1× |

Table 1: **Training time comparison** between a dense encoder (w/ [M]) and a sparse encoder (w/o [M]) in MAE. The encoder is ViT-L (1024-d, 24-block); the decoder is our default (512-d, 4-block). With a masking ratio of 90%, the sparse variant reduces FLOPs by 7.7×. This reduces computation time by 5.8×. In our infra, computation is so fast that data loading becomes a bottleneck, which leads to an actual speedup of 4.1×. Profiling is with synchronized SGD over 16 nodes, each with 8 A100 GPUs and 80 CPU cores. The training length is 800 epochs.

## 5.1 Performance

Fig. 5 compares MAE pre-training *vs.* no pre-training (*i.e.*, training from scratch), using vanilla ViT-L [18]. The from-scratch recipe follows [77] and has 71.4% accuracy.[2] As a comparison, using MAE pre-training for 800 epochs, the same vanilla ViT-L achieves 84.4% accuracy, which has a large increase of **13.0%** absolute *vs.* training from scratch. This gap is much larger than that on image recognition tasks (∼3% [31]), suggesting that MAE pre-training is more helpful for video recognition.

In addition to the accuracy gain, MAE pre-training can *reduce* the overall training cost, as plotted in Fig. 5. The 800-epoch MAE pre-training only takes 35.8 hours. A short fine-tuning (100 epochs here), which takes 16.3 hours, achieves good accuracy thanks to pre-training. The overall training time can be *shorter* than training from scratch (*e.g.*, 400 epochs, 65.2 hours), which converges more slowly without pre-training. This shows that MAE is a practical solution to video recognition.

MAE pre-training is fast because its encoder is only applied on the sparse set of visible patches, without the mask token [M]. We profile the pre-training performance in Table 1. With a masking ratio of 90%, the sparse encoder reduces the FLOPs (floating-point operations) by >10×. After counting the decoder, the sparse design of MAE reduces FLOPs by 7.7×. In our implementation, this reduction should produce a 5.8×computational speedup, if the video data *were* already pre-processed and loaded in memory. Our speedup ratio is *so high* that the video pre-processing and loading time becomes a new bottleneck. In our system, the data loading step increases the wall-clock training time from 24.5 hours to 35.8 hours. Nevertheless, this still leads to a significant speedup of 4.1×.[3]

## 5.2 Ablation experiments

**Masking ratio.** Fig. 6 shows the influence of the masking ratio jointly with the pre-training length. The ratio of 90% works the best. The ratio of 95% performs surprisingly well, which can catch up if trained long enough (Fig. 6 left). A higher masking ratio leads to *fewer* tokens encoded by the encoder; to have a more comprehensive look, we plot the results w.r.t. the total number of encoded tokens (Fig. 6 right). Under this measure, the ratios of 90% and 95% perform closely.

The lower masking ratios of 75% and 50% perform worse, even though the encoder sees more tokens and has higher computation cost. The ratio of 75% is optimal for its image counterpart [31], but not for videos. This observation can be explained by the assumption that video data is more information-redundant.

---

[2]The ViT-B result is 68.5% [77] trained from scratch using this recipe.

[3]The speedup is closer to 5.8× if using *slower* GPUs (V100 instead of A100) that can hide the loading time.

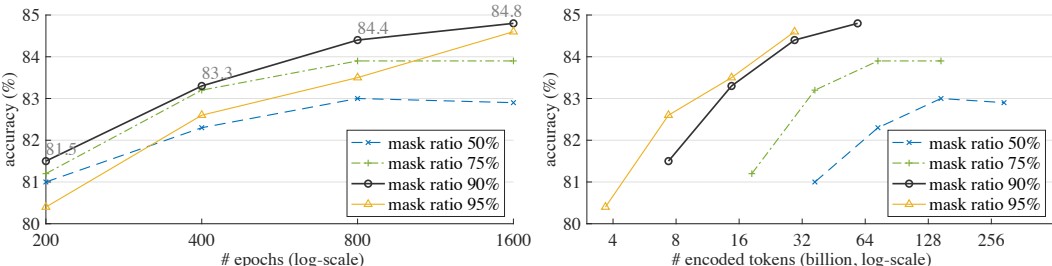

Figure 6: **Masking ratio**. Every point represents a single pre-training and fine-tuning experiment. **Left**: x-axis is the epochs (proportional to the number of *decoded* tokens). **Right**: x-axis is the number of *encoded* tokens.

| case | ratio | acc. |
|---|---|---|
| agnostic | 90 | **84.4** |
| space-only | 90 | 83.5 |
| time-only | 75 | 79.1 |
| block | 75 | 83.2 |

(a) **Mask sampling**. See also Fig. 4. Random sampling that is spacetime-*agnostic* works the best.

| case | acc. |
|---|---|
| pixel (w/o norm) | 83.8 |
| pixel (w/ norm) | **84.4** |
| HOG | 84.0 |
| dVAE token | 83.8 |

(b) **Reconstruction target**. Pixels as reconstruction targets work well with no domain knowledge.

| case | acc. |
|---|---|
| center crop | 83.9 |
| rand crop | **84.4** |
| rand crop (stronger) | 83.4 |
| rand crop + color jit | 83.8 |

(c) **Data augmentation**. Strong augmentation is unnecessary.

| rep. | acc. | speed |
|---|---|---|
| 1 | 83.7 | 1.0× |
| 2 | 84.3 | 1.8× |
| 4 | **84.4** | **3.0×** |

(d) **Repeated sampling**. All entries see the same # samples. Data loading overhead is reduced.

| dim | acc. |
|---|---|
| 128 | 80.8 |
| 256 | 83.1 |
| 512 | **84.4** |
| 1024 | 83.7 |

(e) **Decoder width**. Unlike the image counterpart [31], an overly narrow decoder degrades accuracy noticeably.

| blocks | acc. |
|---|---|
| 1 | 83.2 |
| 2 | 83.6 |
| 4 | **84.4** |
| 8 | 84.3 |

(f) **Decoder depth**. Unlike the image counterpart [31], an overly shallow decoder degrades accuracy.

Table 2: **Ablation experiments** on Kinetics-400. The model is ViT-L, with an input size of $16\times224\times224$ and a spacetime patch size of $2\times16\times16$. The pre-training length is 800 epochs. The entries marked in  gray  are the same, which specify the default settings. This table format follows [31].

**Mask sampling strategy.** Our method follows the structure-agnostic random sampling methodology in BERT [15] and MAE [31]. Table 2a reports that this simple solution works the best in our method.

We compare with other strategies as illustrated in Fig. 4. *Space-only* sampling, which samples on the 2D spatial axes and broadcasts along the temporal axis, works reasonably well (83.5%). *Time-only* sampling, with a masking ratio of 75% (*i.e.*, keep 2 time steps out of 8), performs poorly (79.1%); if we increase its masking ratio to 87.5% (keep 1 out of 8), the accuracy drops further to 75.4%. Time-only sampling is related to future/past frame prediction, which can be an overly difficult task in our scenario. Block-wise sampling [3], in its spacetime variant [77], has 83.2% accuracy with 75% masking ratio (a higher ratio is worse).

**Reconstruction target.** Our method performs decently by reconstructing the original, unmodified pixels (83.8%, Table 2b). Using per-patch normalized pixels [31] improves by 0.6%. This observation is similar to that of its image counterpart [31]. Using HOG [13] as the target [77] works strongly too.

The autoencoding nature of our method (*i.e.*, predicting pixels) provides a self-contained solution. In contrast, an extra tokenizer (*e.g.*, dVAE [47, 9]), as is used in [3, 73], may require external data to train and additional domain knowledge to design (*e.g.*, the dVAE used is a ConvNet [37]). Applying the extra dVAE tokenizer to each frame is computationally heavy, which slows down training by 1.6× in our implementation. Our pixel-based method is simpler and performs better (Table 2b).

**Data augmentation.** Temporal data can provide natural augmentation, *e.g.*, on view points, motion, deformation, occlusion. These forms of natural augmentation have been incorporated by random temporal sampling. Table 2c compares additional augmentation on the spatial domain. Even using *no* spatial augmentation (center crop only) works competitively, similar to the observation on images [31]. Random cropping with a mild scale range of $[0.5, 1]$ works well, while stronger cropping (range $[0.08, 1]$, [63]) reduces accuracy; adding color jittering reduces accuracy too, similar to [31].

| pre-train set | # pre-train data | pre-train method | K400 | AVA | SSv2 |
|---|---|---|---|---|---|
| - | - | none (from scratch) | 71.4 | - | - |
| IN1K | 1.28M | supervised | 78.6 | 17.8 | 50.2 |
| IN1K | 1.28M | MAE | 82.3 | 27.2 | 65.6 |
| K400 | 240k | supervised | - | 22.2 | 55.7 |
| K400 | 240k | MAE | 84.8 | 32.3 | 72.1 |
| K600 | 387k | MAE | **84.9** | 33.7 | 73.0 |
| K700 | 537k | MAE | n/a$^\dagger$ | 34.2 | **73.6** |
| IG-uncurated | 1M | MAE | 84.4 | **35.1** | **73.6** |

Table 3: **Influence of pre-training data**, evaluated on K400, AVA, and SSv2 as the downstream tasks. The MAE pre-training length is 1600 epochs on K400/600/700 and IG-uncurated. No intermediate fine-tuning is used. The model is ViT-L. $^\dagger$: *The K700 training set has 13.9k duplicated videos with the K400 validation set (19.9k), so it is not legitimate to train on K700 to get K400 results.*

It is practically valuable for self-supervised learning methods to be *less dependent* on data augmentation. There are a variety of applications in which augmentation is not valid or is hard to induce, *e.g.*, medical imaging, hyper-spectral imaging, remote sensing, geometric data (point cloud, key points, *etc.*), and their temporal extensions. Our method could be generalized to these cases.

**Repeated sampling.** As our method is fast in computation, we adopt repeated sampling [33] to reduce the data loading overhead. Table 2d reports its influence. Using 2 or 4 repetitions increases wall-clock speed by $1.8\times$ or $3.0\times$, as a loaded and decompressed file is reused multiple times.

**Decoder capacity.** Table 2e and 2f report the influence of the decoder width and depth. Using an overly small decoder degrades accuracy by large margins. This is unlike its image counterpart [31], in which a 128-d or 1-block decoder has no degradation if fine-tuning is applied. We hypothesize that the higher-dimensional video data are more complex and thus require higher decoding capacity. On the other hand, our optimal decoder (512-d, 4-block) is still substantially smaller than the encoder (1024-d, 24-block). This is similar to the observation on its image counterpart [31].

## 5.3 Influence of Data

**Transfer learning ablation.** Table 3 studies pre-training on different datasets and transferring to various downstream tasks. The pre-training datasets include ImageNet-1K (IN1K) [14] and Kinetics-400, 600, and 700 [35, 6, 7]. The downstream tasks include K400, AVA [29], and SomethingSomething v2 (SSv2) [27]. We do *not* perform any intermediate fine-tuning (see appendix), so the comparison here is influenced by the data scale/distribution but not by the number of their labels.

First we compare with pre-training on the IN1K images. MAE pre-training on IN1K[4] is 3.7% better than IN1K supervised pre-training (78.6 to 82.3%); this image-based MAE is even better than K400 *supervised* pre-training, on both AVA (21.6% to 26.3%) and SSv2 (55.7% to 65.6%).

MAE pre-training on K400 has *massive* gains over supervised pre-training on K400: it improves by **10.1**% on AVA (22.2% to 32.3%) and **16.4**% on SSv2 (55.7% to 72.1%). MAE pre-training on K400 videos also substantially outperforms MAE pre-training on IN1K images: it increases by **2.5**% on K400 (82.3% to 84.8%), **5.1**% on AVA (27.2% to 32.3%), and **6.5**% on SSv2 (65.6% to 72.1%), suggesting that MAE pre-training on videos is highly beneficial for these video tasks.

With more pre-training data (K600/K700) without labels, we observe noticeable improvements on AVA and SSv2: comparing with K400 pre-training, MAE with K700 has an extra gain of **1.9**% gain on AVA (32.3% to 34.2%) and **1.5**% on SSv2 (72.1% to 73.6%).

**Real-world data.** We further study MAE pre-training on *real-world* Instagram videos. We study two sets: (i) Instagram videos *curated* (IG-curated) [24] with hashtags similar to K400 classes, and (ii) random, *uncrated* Instagram videos (IG-uncurated). Both sets have 1 million videos.

Table 3 (last row) reports transfer learning results on AVA and SSv2 using IG-*uncurated* pre-training. Notably, on AVA, MAE with IG-uncurated is *better* than MAE with curated Kinetics pre-training (*e.g.*, by **3.1/1.7/1.1**% over K400/600/700 pre-training); on SSv2, MAE with IG-uncurated is among the best, on par with the K700 counterpart.

---

[4]The IN1K pre-trained model is from https://github.com/facebookresearch/mae.

| data | # videos | 200-ep. | 400-ep. | 800-ep. |
|---|---|---|---|---|
| K400 | 240k | 81.5 | 83.3 | **84.4** |
| IG-curated | 240k | 79.0 | 81.6 | 83.2 |
| IG-curated | 512k | 81.9 | 83.5 | 83.9 |
| IG-curated | 1M | **83.5** | 84.1 | 84.2 |
| IG-uncurated | 1M | 83.2 | **84.5** | **84.4** |

Table 4: **Real-world Instagram data** for MAE pre-training. We pre-train MAE on each individual set for 200, 400, and 800 epochs. We compare fine-tuning accuracy on K400. The model is ViT-L.

Table 4 presents more results on the dataset size and training epochs. Pre-training on a 240k subset of IG-curated (the same size as K400) performs worse on K400 classification, which can be caused by the domain shift of data. However, increasing the dataset size of IG-curated to 512k and 1M shows good gains: under the same number of pre-training epochs (200 and 400), it can *outperform* K400 pre-training even when evaluating on K400. IG-uncurated performs similarly well as IG-curated, although the videos are randomly sampled and unrelated to K400 classes. This behavior is *not* observed on contrastive learning methods for videos: *e.g.*, in [22] it is empirically shown that data curation has a major impact on contrastive learning [32, 10, 28] performance.

We believe that our exploration with real-world data has encouraging results. It is a more realistic use case of unsupervised learning at scale. We hope this exploration will shed light on future study.

## 5.4 System-level Comparisons

We provide system-level comparisons with the leading results on K400, AVA, and SSv2. The detailed tables are in the appendix (Table 7, 8, 9). These results are multifaceted, involving architecture designs, computational complexity, model sizes, input resolution, pre-training data and methods, *etc.*, as we summarize in the tables. Our results are competitive and are close to the leading entries. In particular, our results are based only on *vanilla* ViT architectures, while the leading methods are hierarchical or specialized for videos. Our results demonstrate the potential of using fewer inductive biases and learning more from data, which is a pursuit of self-supervised learning.

## 5.5 Video Pre-training for Image Recognition

Finally, we report preliminary results on video pre-training for image recognition. The usage of vanilla ViT allows to convert to 2D easily: we only "deflate" patch embeddings by summing in time. Using ViT-L pre-trained by MAE on K400 / IG-uncurated, we obtain 83.7% / 84.1% accuracy on IN1K image classification. This is better than training ViT-L from scratch on IN1K (82.6% [31]), though lower than MAE pre-training on IN1K (85.9% [31]). Considering the large domain gap, we believe this result is decent and its improvement over training from scratch is encouraging. We hope it will motivate the community to explore video pre-training for *general* visual representation learning.

## 6 Conclusion

We have explored a simple extension of MAE [31] to video data. We have drawn several interesting observations. (i) We find that it is possible to learn strong representations with minimal domain knowledge or inductive biases. This follows the spirit of the ViT paper [18]. Similar to BERT [15] and MAE [31], we show that self-supervised learning on videos can be tackled in a conceptually unified framework. (ii) We empirically show that the masking ratio is an important factor for general masked autoencoding methods [69], and its optimal values may depend on the nature of the data (language, images, videos, *etc.*). (iii) We report encouraging results of pre-training on real-world, uncurated data. It achieves strong performance, close to pre-training on controlled, curated data (*e.g.*, Kinetics). To the best of our knowledge, promising results on uncurated data are rare in the literature.

In spite of these observations, open problems remain. The scale of data we have explored is orders of magnitudes smaller than the language counterparts [52, 15, 53, 5]. While our method has largely improved the efficiency of self-supervised learning, the high-dimensional video data still present a major challenge for scaling up. We hope our study will provide initial signals for future research.

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
