# A Implementation Details

**Kinetics action classification.** Our settings mainly follow [31, 77]. Table 5a summarizes our pre-training settings on Kinetics. Table 5b shows the corresponding fine-tuning settings for ViT-B/L/H. For fine-tuning, we add a linear classifier layer to the encoder's averaged tokens [18].

For fine-tuning the intermediately fine-tuned checkpoints from K600 in Table 7, we use the setting in Table 5b with a lower learning rate (8e-4) and shorter duration (40 epochs for ViT-L; 30 for ViT-H) and an increased drop path rate of 0.3 for ViT-H.

**AVA action detection.** Table 6a summarizes our fine-tuning settings on AVA [29]. The settings mainly follow [39, 77]. We follow the detection architecture in [21, 39, 77] that adapts Faster R-CNN [57] for video action detection. Only for the AVA results in Table 8, we use relative positions [59, 54] (as implemented in [39]) during fine-tuning.

**SSv2 action classification.** Table 6b summarizes our fine-tuning settings on SSv2 [27]. The settings mainly follow [39, 77]. For the frame sampling, we split each video into segments, and sample one frame from each segment to form a clip following [39, 19].

**Fine-tuning from image pre-training.** In Table 3 we have compared with ImageNet-based supervised/MAE pre-training. When fine-tuning these variants for videos, we inflate the 2D kernel of the patch embedding layer to 3D [8] and initialize the temporal position embeddings by zero.

| config | value |
|---|---|
| optimizer | AdamW [43] |
| optimizer momentum | $\beta_1, \beta_2{=}0.9, 0.95$ [9] |
| weight decay | 0.05 |
| learning rate | 1.6e-3 |
| learning rate schedule | cosine decay [42] |
| warmup epochs [26] | 120 |
| epochs | default 800 |
| repeated sampling [33] | 4 |
| augmentation | hflip, crop [0.5, 1] |
| batch size | 512 |
| gradient clipping | 0.02 |

(a) Kinetics pre-training

| config | ViT-B | ViT-L | ViT-H |
|---|---|---|---|
| optimizer | | AdamW [43] | |
| optimizer momentum | | $\beta_1, \beta_2{=}0.9, 0.999$ | |
| weight decay | | 0.05 | |
| learning rate | 1.6e-2 | 4.8e-3 | 1.6e-3 |
| learning rate schedule | | cosine decay [42] | |
| warmup epochs [26] | | 5 | |
| epochs | 150 | 100 | 75 |
| repeated sampling [33] | 2 | 2 | 1 |
| augmentation | | RandAug (9, 0.5) [12] | |
| batch size | 768 | 256 | 256 |
| mixup [86] | | 0.8 | |
| cutmix [84] | | 1.0 | |
| label smoothing [64] | | 0.1 | |
| drop path [34] | 0.1 | 0.2 | 0.2 |
| dropout [60] | 0.3 | 0.3 | 0.5 |
| layer-wise decay [11] | 0.65 | 0.75 | 0.8 |

(b) Kinetics fine-tuning

Table 5: Settings on Kinetics.

| config | values |
|---|---|
| optimizer | SGD |
| weight decay | 1e-8 |
| learning rate | 7.2(L), 4.8(H) |
| learning rate schedule | cosine decay [42] |
| warmup epochs [26] | 5 |
| epochs | 30 |
| batch size | 128 |
| drop path [34] | 0.2 |
| dropout [60] | 0.5 |
| layer-wise decay [11] | 0.75 (L) 0.85 (H) |

(a) AVA fine-tuning

| config | values |
|---|---|
| optimizer | SGD |
| weight decay | 1e-4 |
| learning rate | 0.64 (L) 0.32 (H) |
| learning rate schedule | cosine decay [42] |
| warmup epochs [26] | 3 |
| epochs | 40 |
| augmentation | RandAug (9, 0.5) [12] |
| batch size | 256 |
| mixup [86] | 0.8 |
| cutmix [84] | 1.0 |
| label smoothing [64] | 0.1 |
| drop path [34] | 0.2 |
| dropout [60] | 0.5 |
| layer-wise decay [11] | 0.75 (L) 0.85 (H) |

(b) SSv2 fine-tuning

Table 6: Settings on AVA and SSv2. (L) and (H) stands for ViT-L and ViT-H, respectively.

| pre-train | extra data | architecture | input size | top-1 | top-5 | FLOPs | param. |
|---|---|---|---|---|---|---|---|
| scratch | | SlowFast [21] | $64 \times 224^2$ | 79.8 | 93.9 | $234 \times 3 \times 10$ | 60 |
| scratch | | X3D-XL [20] | $16 \times 312^2$ | 79.1 | 93.9 | $48 \times 3 \times 10$ | 11 |
| scratch | | MoViNet [36] | $120 \times 320^2$ | 81.5 | 95.3 | $386 \times 1 \times\ \ 1$ | 31 |
| scratch | | MViT-B [19] | $64 \times 224^2$ | 81.2 | 95.1 | $455 \times 3 \times\ \ 3$ | 37 |
| scratch | | MViTv2-B [19] | $32 \times 224^2$ | 82.9 | 95.7 | $255 \times 1 \times\ \ 5$ | 51 |
| supervised | IN21K | Swin-B [41] | $32 \times 224^2$ | 82.7 | 95.5 | $282 \times 3 \times\ \ 4$ | 88 |
| supervised | IN21K | Swin-L [41] | $32 \times 224^2$ | 83.1 | 95.9 | $604 \times 3 \times\ \ 4$ | 197 |
| supervised | IN21K | Swin-L [41] | $32 \times 384^2$ | 84.9 | 96.7 | $2107 \times 5 \times 10$ | 200 |
| BEVT [73] | IN1K+DALLE | Swin-B [41] | $32 \times 224^2$ | 81.1 | n/a | $282 \times 3 \times\ \ 4$ | 88 |
| MaskFeat [77] | | MViTv2-L [39] | $16 \times 224^2$ | 84.3 | 96.3 | $377 \times 1 \times 10$ | 218 |
| MaskFeat [77] | | MViTv2-L [39] | $40 \times 352^2$ | 86.7 | 97.3 | $3790 \times 3 \times\ \ 4$ | 218 |
| MaskFeat [77] | K600 | MViTv2-L [39] | $40 \times 352^2$ | 87.0 | 97.4 | $3790 \times 3 \times\ \ 4$ | 218 |
| **MAE** | | ViT-B | $16 \times 224^2$ | 81.3 | 94.9 | $180 \times 3 \times\ \ 7$ | 87 |
| **MAE** | | ViT-L | $16 \times 224^2$ | 84.8 | 96.2 | $598 \times 3 \times\ \ 7$ | 304 |
| **MAE** | | ViT-H | $16 \times 224^2$ | 85.1 | 96.6 | $1193 \times 3 \times\ \ 7$ | 632 |
| **MAE** | | ViT-L | $40 \times 312^2$ | 85.8 | 96.9 | $4757 \times 3 \times\ \ 7$ | 304 |
| **MAE** | | ViT-H | $32 \times 312^2$ | 86.0 | 97.0 | $6382 \times 3 \times\ \ 7$ | 632 |
| **MAE** | K600 | ViT-L | $16 \times 224^2$ | 86.5 | **97.2** | $598 \times 3 \times\ \ 7$ | 304 |
| **MAE** | K600 | ViT-H | $16 \times 224^2$ | **86.8** | **97.2** | $1193 \times 3 \times\ \ 7$ | 632 |
| *using in-house data for supervision:* | | | | | | | |
| supervised | JFT-300M | ViViT-L [2] | $32 \times 320^2$ | 83.5 | 94.3 | $3980 \times 3 \times\ \ 1$ | 308 |
| supervised | JFT-300M | ViViT-H [2] | $32 \times 320^2$ | 84.9 | 95.8 | $3981 \times 3 \times\ \ 4$ | 654 |
| supervised + text | FLD-900M | Florence [83] | $n/a \times 384^2$ | 86.5 | 97.3 | $n/a \times 3 \times\ \ 4$ | 647 |
| SimMIM [80] + sup. | IN21K+70M | SwinV2-G [40] | $8 \times 384^2$ | 86.8 | n/a | $n/a \times 5 \times\ \ 4$ | 3000 |
| supervised | JFT-3B+SSv2+MiT+IN | CoVeR [85] | $16 \times 448^2$ | 87.2 | n/a | $n/a \times 3 \times\ \ 1$ | n/a |
| supervised | WTS-60M | MTV-H [82] | $32 \times 280^2$ | 89.9 | 98.3 | $6130 \times 3 \times\ \ 4$ | n/a |

Table 7: **System-level comparisons on Kinetics-400 action classification**. We report top-1 and top-5 accuracy on the validation set. The input size is $T \times H \times W$. FLOPs (in $10^9$) are presented as "FLOPs per view $\times$ spatial views $\times$ temporal views", following the literature. Parameters are in $10^6$. The "extra data" column specifies the data used in addition to K400. Entries using spatial resolution $>224^2$ are noted in gray; entries using in-house data for supervision are in light blue. Our results with K600 are with intermediate fine-tuning.

*This table does not include results using K700, because the K700 training set has 13.9k videos duplicated with the K400 validation set (19.9k). Results with K700 are in Table 8 (AVA) and Table 9 (SSv2).*

# B  Additional Experimental Results

## B.1  System-level Comparisons

**Kinetics-400.** Table 7 compares on Kinetics-400 (K400). Our results are competitive with the leading ones. Importantly, our method is much *simpler* than many other entries. Our method is the only leading entry based on *vanilla* ViT, while others were based on hierarchical or specialized designs for videos. Our model does *not* use relative position embedding, which could have extra gains that are orthogonal to our thesis. Our results can compete with some strong results that were based on in-house data for supervision. Our models achieve this at standard 224×224 spatial resolution, while higher-resolution fine-tuning and testing may improve results at a higher cost, as shown in gray indicating entries using spatial resolution $>224^2$.

**AVA.** Table 8 compares on AVA [29] action detection. Using only a resolution of $16 \times 224^2$, our results are close to those of MaskFeat on higher-resolution inputs ($40 \times 312^2$). Importantly, our architectures are plain ViT models without feature hierarchies, yet they perform strongly on this detection task.

**SSv2.** Table 9 compares on SSv2 [27] action classification. On the resolution of $16 \times 224^2$ and using vanilla ViT, our results compare favorably with those of MaskFeat on $40 \times 312^2$ inputs.

| pre-train | pre-train data | architecture | input size | mAP center | mAP full | FLOPs | param. |
|---|---|---|---|---|---|---|---|
| supervised | K400 | SlowFast [21] | $32 \times 224^2$ | 23.8 | - | 138 | 53 |
| supervised | K400 | MViTv1-B [19] | $64 \times 224^2$ | 27.3 | - | 455 | 36 |
| supervised | K400 | MViTv2-B [39] | $32 \times 224^2$ | 28.1 | 29.0 | 225 | 51 |
| MaskFeat [77] | K400 | MViTv2-L [39] | $40 \times 312^2$ | **36.3** | **37.5** | 2828 | 218 |
| **MAE** | K400 | ViT-L | $16 \times 224^2$ | 35.9 | 36.8 | 598 | 304 |
| **MAE** | K400 | ViT-H | $16 \times 224^2$ | **36.8** | **37.4** | 1193 | 632 |

(a) **AVA results using Kinetics-400 pre-training**

| pre-train | pre-train data | architecture | input size | mAP center | mAP full | FLOPs | param. |
|---|---|---|---|---|---|---|---|
| supervised | K600 | SlowFast [21] | $64 \times 224^2$ | 27.5 | - | 296 | 59 |
| supervised | K600 | X3D-XL [20] | $16 \times 312^2$ | 27.4 | - | 48 | 11 |
| supervised | K600 | MViT-B [19] | $32 \times 224^2$ | 28.7 | - | 236 | 53 |
| supervised | K600 | MViTv2-B [39] | $32 \times 224^2$ | 29.9 | 30.5 | 225 | 51 |
| supervised | K600 | ACAR [48] | $64 \times 224^2$ | - | 31.4 | n/a | n/a |
| MaskFeat [77] | K600 | MViTv2-L [39] | $40 \times 312^2$ | **37.8** | **38.8** | 2828 | 218 |
| **MAE** | K600 | ViT-L | $16 \times 224^2$ | 37.7 | 38.4 | 598 | 304 |
| **MAE** | K600 | ViT-H | $16 \times 224^2$ | **39.2** | **40.3** | 1193 | 632 |

(b) **AVA results using Kinetics-600 pre-training**

| pre-train | pre-train data | architecture | input size | mAP center | mAP full | FLOPs | param. |
|---|---|---|---|---|---|---|---|
| supervised | K700 | MViTv2-B [39] | $32 \times 224^2$ | 31.3 | 32.3 | 225 | 51 |
| supervised | K700 | ACAR [48] | $64 \times 224^2$ | - | 33.3 | n/a | n/a |
| supervised | K700 + IN21K | MViTv2-L [39] | $40 \times 312^2$ | 33.5 | 34.4 | 2828 | 213 |
| **MAE** | K700 | ViT-L | $16 \times 224^2$ | 38.4 | 39.5 | 598 | 304 |
| **MAE** | K700 | ViT-H | $16 \times 224^2$ | **39.3** | **40.1** | 1193 | 632 |

(c) **AVA results using Kinetics-700 pre-training**

Table 8: **System-level comparisons on AVA v2.2 action detection**. We report mAP using center-crop or full-resolution inference, following the literature. FLOPs (in $10^9$) are measured with center-crop inference. Parameter numbers are in $10^6$. Only in this table, following MaskFeat [77], our results are with intermediate fine-tuning and with relative positions [59, 54] during fine-tuning.

| pre-train | pre-train data | architecture | input size | top-1 | top-5 | FLOPs | param. |
|---|---|---|---|---|---|---|---|
| supervised | K400 | SlowFast [21] | $32 \times 224^2$ | 63.1 | 87.6 | $106 \times 3 \times 1$ | 53 |
| supervised | K400 | MViTv1-B [19] | $64 \times 224^2$ | 67.7 | 90.9 | $454 \times 3 \times 1$ | 37 |
| supervised | K400 | MViTv2-B [39] | $32 \times 224^2$ | 70.5 | 92.7 | $225 \times 3 \times 1$ | 51 |
| supervised | K400 + IN21K | Swin-B [41] | $32 \times 224^2$ | 69.6 | 92.7 | $321 \times 3 \times 1$ | 89 |
| supervised | K400 + IN21K | MViTv2-B [39] | $32 \times 224^2$ | 72.1 | 93.4 | $225 \times 3 \times 1$ | 51 |
| supervised | K400 + IN21K | MViTv2-L [39] | $40 \times 224^2$ | 73.3 | 94.1 | $2828 \times 3 \times 1$ | 213 |
| BEVT [73] | K400 + IN1K | Swin-B [41] | $32 \times 224^2$ | 71.4 | n/a | $321 \times 3 \times 1$ | 88 |
| MaskFeat [77] | K400 | MViTv2-L [39] | $40 \times 312^2$ | **74.4** | **94.6** | $2828 \times 3 \times 1$ | 218 |
| **MAE** | K400 | ViT-L | $16 \times 224^2$ | 72.1 | 93.9 | $598 \times 3 \times 1$ | 304 |
| **MAE** | K400 | ViT-H | $16 \times 224^2$ | **74.1** | **94.5** | $1193 \times 3 \times 1$ | 632 |

(a) **SSv2 results using Kinetics-400 pre-training**

| pre-train | pre-train data | architecture | input size | top-1 | top-5 | FLOPs | param. |
|---|---|---|---|---|---|---|---|
| supervised | K600 | MViTv1-B [19] | $32 \times 224^2$ | 67.7 | 90.9 | $454 \times 3 \times 1$ | 37 |
| MaskFeat [77] | K600 | MViTv2-L [39] | $40 \times 312^2$ | **75.0** | **95.0** | $2828 \times 3 \times 1$ | 218 |
| **MAE** | K600 | ViT-L | $16 \times 224^2$ | 73.0 | 94.2 | $598 \times 3 \times 1$ | 304 |
| **MAE** | K600 | ViT-H | $16 \times 224^2$ | **75.2** | **94.9** | $1193 \times 3 \times 1$ | 632 |

(b) **SSv2 results using Kinetics-600 pre-training**

| pre-train | pre-train data | architecture | input size | top-1 | top-5 | FLOPs | param. |
|---|---|---|---|---|---|---|---|
| **MAE** | K700 | ViT-L | $16 \times 224^2$ | 73.6 | 94.4 | $598 \times 3 \times 1$ | 304 |
| **MAE** | K700 | ViT-H | $16 \times 224^2$ | **75.5** | **95.0** | $1193 \times 3 \times 1$ | 632 |

(c) **SSv2 results using Kinetics-700 pre-training**

Table 9: **System-level comparisons on SSv2 action classification**. Notations of FLOPs ($10^9$) and parameters ($10^6$) follow Table 7. We do not use intermediate fine-tuning here (see Table 10).

## B.2   Ablation on Intermediate Fine-tuning

In Table 3 we have shown results of self-supervised pre-training directly transferred to downstream datasets. Following the literature, we also investigate an another scenario: after self-supervised pre-training, we perform *intermediate fine-tuning* on the pre-training set using labels, before transferring. Table 10 studies its influence. Intermediate fine-tuning has substantial improvements on AVA, while on SSV2 its effect is marginal.

| pre-train data | # | intermediate FT | K400 | AVA | SSv2 |
|---|---|---|---|---|---|
| K400 | 240k | | 84.8 | 32.3 | 72.1 |
| K400 | 240k | ✓ | - | 36.8 | 72.6 |
| K600 | 387k | | 84.9 | 33.7 | 73.0 |
| K600 | 387k | ✓ | 86.5 | 37.9 | 73.1 |
| K700 | 537k | | n/a | 34.2 | 73.6 |
| K700 | 537k | ✓ | n/a | 39.3 | 73.7 |

Table 10: **Influence of intermediate fine-tuning**, evaluated on AVA and SSv2. The model is ViT-L. The MAE pre-training length is 1600 epochs on K400/600/700. Using K700 training set for K400 validation is not legitimate due to the duplications in these training and validation sets.

## B.3   Masking during fine-tuning

We perform an ablation that applies masking during the supervised fine-tuning phase. We explore a masking ratio of 50% that is annealed to 0% with a cosine schedule during fine-tuning. The result is 84.1%, compared to 84.4% for full fine-tuning without masking, but at a 1.2× speedup. If we start fine-tuning with a masking ratio of 50% and anneal it to 0%, the accuracy is 83.8% at a speedup of 1.3×. The experiments are summarized in Table 11. We think this is an interesting result showing that masking can also speedup fine-tuning.

| start fine-tune masking ratio | K400 accuracy | speed |
|---|---|---|
| 0% | 84.4 | 1.0× |
| 50% | 84.1 | 1.2× |
| 75% | 83.8 | 1.3× |

Table 11: **Masking during fine-tuning** on Kinetics-400. We use Cosine annealing of masking ratio during fine-tuning. The starting masking ratio is varied between 0% (baseline without masking), 50% and 75%. The annealing is towards 0% at the end of fine-tuning. The model is ViT-L and the MAE pre-training length is 800 epochs on K400; *cf*. Table 2.

## B.4   Ablation on SSv2

We perform a subset of the ablations that were carried out for Kinetics in Table 2 on the SSv2 dataset. We directly pre-train and fine-tune on SSv2 and use a short pre-training schedule of 200 epochs to save training resources. The results in Table 12 indicate that the default choices for Kinetics also lead to good performance on SSv2. Namely, spacetime agnostic mask sampling (Table 12a) as well as decoder width (12b) of 512 and depth (12c) of 4 provide better accuracy than other design choices.

| case | ratio | acc. |
|---|---|---|
| agnostic | 90 | **63.4** |
| space-only | 90 | 59.5 |
| time-only | 75 | 61.9 |

| dim | acc. |
|---|---|
| 128 | 59.4 |
| 256 | 63.2 |
| 512 | **63.4** |

| blocks | acc. |
|---|---|
| 1 | 63.9 |
| 2 | **63.4** |
| 4 | **63.4** |
| 8 | 62.0 |

(a) **Mask sampling**. See also Fig. 4 and Table 2. Random sampling that is spacetime-*agnostic* works best.

(b) **Decoder width**. Similar to Table 2, a narrow decoder (128-d) drops accuracy.

(c) **Decoder depth**. Four or two decoder layers provides good accuracy on SSv2.

Table 12: **Ablation experiments** on SSv2. We use a short pre-training length of 200 epochs. The model is ViT-L, with an input size of 16×224×224 and a spacetime patch size of 2×16×16. This table format follows [31] and Table 2. The entries marked in `gray` are the same, which specify the default settings, and achieve best performance (similar to the results for Kinetics in Table 2).

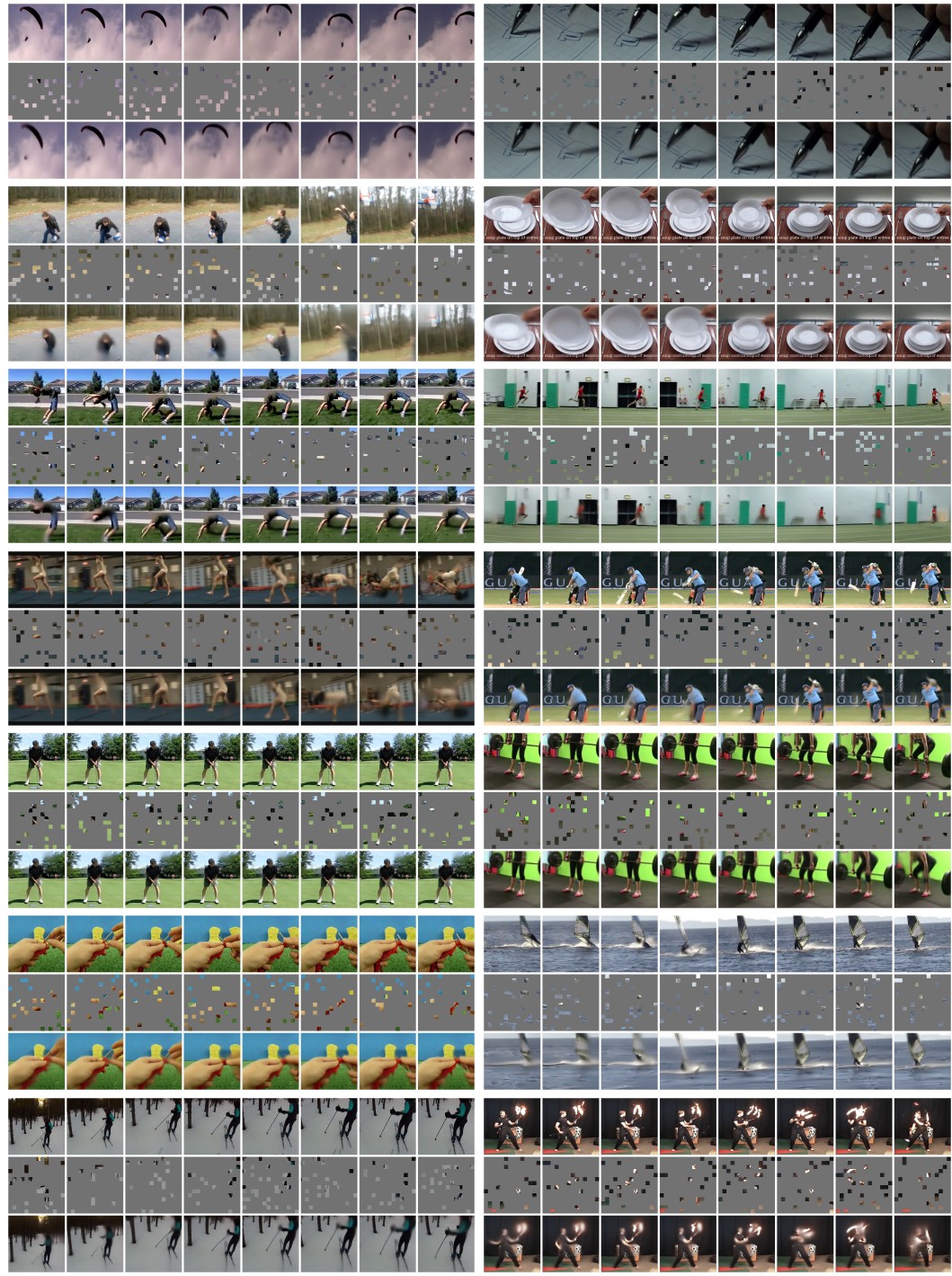

Figure 7: More visualizations on Kinetics-400 following Fig. 2 (masking ratio 90%).

# Acknowledgements

We would like to thank Chen Wei, Karttikeya Mangalam, Chao-Yuan Wu, Ross Girshick, Piotr Dollár, and Jitendra Malik for discussions and feedback.

## C   Potential negative societal impacts

The potential negative societal impact of our approach is related to other machine learning methods that are learning from training data and therefore reflect statistics and biases of the datasets used. Our method could also be used to generate content that may or may not reflect biases of the training data.

Video understanding research requires compute-intensive experiments because of the large input dimensionality and the large training datasets which can have negative environmental impact. To mitigate this effect, our approach greatly reduces the computational cost for performing research. Further, there is potential of misuse for video classification methods, *e.g.* for surveillance purpose.