# OpenReview forum: "Masked Autoencoders As Spatiotemporal Learners"
_NeurIPS.cc/2022/Conference — NeurIPS 2022 Accept_

### Official Review · Reviewer_VF6m · 2022-06-30

**Rating:** 7
**Confidence:** 5
**Soundness:** 3 good
**Presentation:** 4 excellent
**Contribution:** 3 good

**Summary:**

This paper studies a simple extension of image MAE to video domain. The experiments are conducted on a set of standard video datasets, i.e., Kinetics, AVA, SSv2, etc., showing the proposed approach achieve appealing results. This paper also presents some interesting findings: (a) due to high information redundancy, video MAE requires higher mask ratio comparing to its image counterpart; (b) random masking works surprisingly well; (c) architectural findings such as video MAE requires a deeper decoder and a larger decoder hidden size. On the one hand, the ablation experiments are comprehensive which covers many aspects of the model design. On the other hand, these experiments are only conducted on a single dataset, K400, which is not convincing.  As the paper is mainly an analysis paper, instead of proposing a novel approach, the ablation studies are insufficient.

**Questions:**

- L192-194, Fig 6 (right). How do you vary the #encoded tokens when the mask ratio is fixed? By training longer?
- L129 mentions it is sufficient to predict a single time slice (16x16) instead of the full space-time patch ($t$x16x16). Is there any experimental results that compares the two strategies?

**Ethics Review Area:**

["Inadequate Data and Algorithm Evaluation"]

**Limitations:**

Yes.

**Strengths And Weaknesses:**

Strengths

- Simple, yet effective method for video self-supervised learning.
- Comprehensive experimental comparison shows that the proposed approach has strong performance compare to prior work.
- The paper is well-organized and is easy to follow. Most of the claims are supported with solid experimental results.

Weaknesses:

- The ablation experiments in Table 2 are only performed on a single dataset (K400), which may not provide a precise or comprehensive evidence of how these ablated items may affect the model performance. In TimeSformer [4] Table 1, it is shown that the conclusions aren’t exactly the same when looking at results from K400 and SSv2, as K400 is more static while SSv2 requires more temporal modeling. I strongly encourage the authors to also conduct ablation experiments on SSv2 to have more persuasive conclusions.
- Overall the novelty is low, as it is almost a direct extension of image MAE. While there are few interesting conclusions/findings, e.g., (a) video modeling requires a higher mask ratio. (b) random masking works better than structual masking, and (c) decoder in video MAE needs to be more powerful, etc. But these are somewhat expected. I do value the experimental results that validate these findings, but the overall contribution is limited.

---

> ### Author Response · Authors · 2022-08-02
> **Response to Reviewer VF6m**
>
> We thank the reviewer for the feedback and positive comments. Below are our responses.
>
> > The ablation experiments in Table 2 are only performed on a single dataset (K400), which may not provide a precise or comprehensive evidence of how these ablated items may affect the model performance.
>
> We agree that ablation studies on multiple datasets would be better; however, please note that ablations on a single (large) dataset is a standard approach in most papers (that are commonly accepted by top venues) and it is rare to find video classification publications that perform ablations on multiple datasets. Especially since K400 is a large-scale dataset and therefore each ablation is computationally expensive (e.g., the K400 training set has ~72x more frames than ImageNet has images). Please note that we trained representations on 4 datasets: K400 / K600 / K700 / Instagram and evaluated them on 3 dataset: K400, AVA and SSv2 with comparable performance to state-of-the-art on all. We nevertheless agree with the reviewer about properties in SSv2 and have already started to perform a set of main SSv2 ablations that will be included in the final version.
>
> > Overall the novelty is low, as it is almost a direct extension of image MAE. While there are few interesting conclusions/findings, e.g., (a) video modeling requires a higher mask ratio. (b) random masking works better than structual masking, and (c) decoder in video MAE needs to be more powerful, etc. But these are somewhat expected. I do value the experimental results that validate these findings, but the overall contribution is limited.
>
> We fully agree that our work's strength is not in technical novelty. We appreciate that all reviewers are open-minded and have recognized our work's value in many other aspects.
>
> We believe that the scope of novelty is broad and is beyond technical algorithms. It can also include extending existing paradigms to new problems, designing new experiments/ablations, drawing new observations/insights, and verifying expected hypothesis in new problems or tasks. From the reviewer comments, we have seen that many of these values of our work are recognized. We thank the reviewer for seeing value in our work from that perspective.
>
> >L192-194, Fig 6 (right). How do you vary the #encoded tokens when the mask ratio is fixed? By training longer?
>
> This is correct, training is over different number of epochs. The datapoints in Fig. 6 right and left are identical, only the x-axis changes. We will make this more clear.
>
> >L129 mentions it is sufficient to predict a single time slice (16x16) instead of the full space-time patch (x16x16). Is there any experimental results that compares the two strategies?
>
> Yes, predicting the full spacetime patch (2$\times$16$\times$16) has 84.0% accuracy, instead of a single slice (1$\times$16$\times$16) that has 84.4% accuracy. The experimental setting is identical to our baseline in Table 2 of the main paper.

---

> > ### Comment · Reviewer_VF6m · 2022-08-08
> > **Thanks the authors for the response.**
> >
> > Most of my concerns are addressed. I would like to raise my rating from weakly accept to accept.

---

> > > ### Author Response · Authors · 2022-08-09
> > > **Response to Reviewer VF6m**
> > >
> > > We thank the reviewer for the feedback and are glad that most of the concerns are addressed.

---

### Official Review · Reviewer_j1gc · 2022-07-12

**Rating:** 8
**Confidence:** 4
**Soundness:** 4 excellent
**Presentation:** 4 excellent
**Contribution:** 3 good

**Summary:**

This paper proposes to extend MAE for videos. Extensive ablations show that MAE can be successfully extended to videos and the optimal masking ratio is higher than images. It achieves competitive results on popular video classification and detection datasets based on both large uncurated and well-aligned upstream datasets.

**Questions:**

1. Line 162: "it takes K temporal clips (by default K=7 on Kinetics)", any intuition why choose 7-view here?

2. Line 130-131:  "In practice, we find it sufficient to predict a single time slice of the patch (16x16)", which time index?

3. In Table 4, VideoMAE pretrained with IG-uncurated doesn't show big improvement over K400 when both are fine-tuned on in-domain K400. This raises a natural question in my mind: in MAE, does the accuracy continue to significantly improve with a bigger scale of data?

4. A similar question regarding resolution is raised in Table 7,8,9: If VideoMAE with a larger resolution continues to show further gains?

5. Random shuffle, slicing and reindexing causes randomly accessing the memory, which recomputes the cache. In practice, depending on the actual hardware, it may slow down the process (i.e. 90% cannot lead to 10x speed-up). Can authors share any knowledge about it?

**Limitations:**

Yes. The authors have adequately addressed the limitations and potential negative societal impact of their work

**Strengths And Weaknesses:**

1. Simple and effective idea: simple random masking with pixel-level reconstruction works well for videos.

2. Clean design: Standard ViTs with a lightweight decoder is enough to achieve competitive results on video classification and detection.

3. Comprehensive ablation studies on upstream/downstream data, decoder design choices, masking ratio/strategy, and  reconstruction targets.

---

> ### Author Response · Authors · 2022-08-02
> **Response to Reviewer j1gc**
>
> We thank the reviewer for the feedback and positive comments. Below are our responses.
>
> >Line 162: "it takes K temporal clips (by default K=7 on Kinetics)", any intuition why choose 7-view here?
>
> The videos have a duration of 10 seconds with a sampling rate of 30 fps (i.e. 300 frames). Since we sample clips of 16 frames with a sampling stride of 4 (covering a window of 15*4=60 frames), taking 7 clips, uniformly over time, will cover the full video with some overlap between the clips, which works well empirically; e.g. the ViT-L model in Table 7 has 84.5% with 5-clip, 84.8% with 7-clip and 84.8% with 10-clip testing.
>
> > In Table 4, VideoMAE pretrained with IG-uncurated doesn't show big improvement over K400 when both are fine-tuned on in-domain K400. This raises a natural question in my mind: in MAE, does the accuracy continue to significantly improve with a bigger scale of data?
>
> We also think that IG-uncurated performing on-par with the in-domain K400 data could be related to domain gap, and think the result is still encouraging as the data is purely random Instagram videos. For scaling pre-training data, the experiments on AVA in Table 3 show a clearer trend: From K400->K600->K700->IG-uncurated the gains are: +1.4 / +2.0 / +3.1 mAP. Here, IG-uncurated outperforms K400 significantly (+3.1 mAP) on AVA. Since both Kinetics and AVA are human action recogntition datasets, there is more domain overlap between Kinetics and AVA. than random IG videos. So the experiment shows that larger data (K400/K600/K700) improves accuracy, even it is not domain-specific (IG-uncruated). In future work, we hope to explore even larger data scales, beyond 1M videos.
>
> > A similar question regarding resolution is raised in Table 7,8,9: If VideoMAE with a larger resolution continues to show further gains?
>
> Yes, this is an interesting question, increasing the resolution can further improve accuracy:
> We fine-tunied ViT-H, 16$\times$224$\times$224 resolution in Table 7, for 30 epochs with a resolution of 32$\times$312$\times$312 (maximum resolution that fits into memory), and the accuracy increases from 85.1% to 86.1%.
> Furthermore, we have fine-tuned K400 pre-trained ViT-L, 16x224x224 resolution in Table 7, with a resolution of 40$\times$312$\times$312, and the accuracy increases from 84.8% to 85.8%. The experiments are summarized below and we will add these to the final version of the paper.
>
> | architecture | input size     | K400 accuracy |
> | ----------- | ----------- | ----------- |
> | ViT-L | 16$\times$224$^2$     | 84.8%       |
> | ViT-L | 40$\times$312$\times$312     | **85.8**%       |
> | ViT-H | 16$\times$224$^2$     | 85.1%       |
> | ViT-H | 32$\times$312$\times$312     | **86.1**%       |
>
> Table B: Higher resolution results on Kinetics-400. _Cf_. Table 7 of the main paper.
>
> >Random shuffle, slicing and reindexing causes randomly accessing the memory, which recomputes the cache. In practice, depending on the actual hardware, it may slow down the process (i.e. 90% cannot lead to 10x speed-up). Can authors share any knowledge about it?
>
> We do not observe noticeable overhead caused by random shuffling. It can be simply implemented by gather (in PyTorch) or einsum with one-hot indexes (in PyTorch/TensorFlow/JAX), and einsum (which self-attention is based on) is highly optimized in speed.
>
> The <10$\times$ speed-up ratio in 90% masking is not caused by random shuffling. A major overhead is the presence of the decoder which is on all tokens. So even the theoretical speedup is <10$\times$ (7.7$\times$). In practice, the actual speedup is smaller than 7.7$\times$, because smaller computations (e.g., fewer tokens in our case) are less parallelism-friendly for hardware.

---

> > ### Comment · Reviewer_j1gc · 2022-08-08
> > **Thanks for the response**
> >
> > Thanks for the response!  I have updated my scores accordingly. Can author[s] share some knowledge about Q2?

---

> > > ### Author Response · Authors · 2022-08-08
> > > **Response to Reviewer j1gc**
> > >
> > > Thank you. Apologies for not including our response earlier. Please see below and let us know if more information is required.
> > >
> > > > Line 130-131: "In practice, we find it sufficient to predict a single time slice of the patch (16x16)", which time index?
> > >
> > > We predict the 1st time-slice for each patch of size (2x16x16). So for 16 frames, we make 8 predictions of size (1x16x16). Predicting all 16 time-slices leads to slightly lower accuracy (-0.4%).

---

> > > > ### Comment · Reviewer_j1gc · 2022-08-08
> > > > **Thanks for the quick reply**
> > > >
> > > > Interesting. Does it work by removing the temporal tokenization (e.g. keep k_t=1), and predicting the time-slices based on the sampling stride. This removes one variable and could generalize better when evaluated on more diverse datasets.
> > > >
> > > > Lastly, I'm looking forward to seeing the power of MAE in larger data scales!

---

> > > > > ### Author Response · Authors · 2022-08-09
> > > > > **Response to Reviewer j1gc**
> > > > >
> > > > > > Interesting. Does it work by removing the temporal tokenization (e.g. keep k_t=1), and predicting the time-slices based on the sampling stride. This removes one variable and could generalize better when evaluated on more diverse datasets.
> > > > >
> > > > > This is an interesting hypothesis, and could further improve the encouraging video pre-training for image recognition results in Section 5.5 (since deflation is no longer needed). We have not tried this idea yet, but will follow the reviewers' suggestion and add changing the tokenization in this way to the paper. We thank the reviewer for the feedback.

---

### Official Review · Reviewer_esSt · 2022-07-12

**Rating:** 7
**Confidence:** 4
**Soundness:** 4 excellent
**Presentation:** 4 excellent
**Contribution:** 3 good

**Summary:**

The paper extends the recently proposed MAE pipeline for images to the video domain. The method shows competitive performance on video datasets despite having minimum domain knowledge.

**Questions:**

1) It would be very interesting to see qualitative results for future predictions (when every token is masked in several last frames).
2) Did you try to use masking while finetuning for a supervised task? (to obtain speed-up in computation on this stage as well). Maybe one can anneal mask ration in finetuning stage to obtain the best metrics with minimum computational resources.
3) Is it possible to do Gibbs sampling with your pipeline? If yes -- did you try it and what is the quality of the samples?

**Limitations:**

Yes

**Strengths And Weaknesses:**

Strengths

- Simple pipeline with competitive performance
- Great ablation study
- interesting possibility for computational speed-up (as the encoder is only applied on the sparse set of visible patches)
- good performance on the uncurated dataset
- good comparison with other video pretraining methods

Weaknesses
- I don’t see any

Although the novelty seems limited to me as it is a straightforward generalization of MAE method to the video domain, in other aspects the paper seems seamless to me, the work has practical value to the community, and the experiments section is great.

---

> ### Author Response · Authors · 2022-08-02
> **Response to Reviewer esSt**
>
> We thank the reviewer for the feedback and positive comments. Below are our responses.
>
> >It would be very interesting to see qualitative results for future predictions (when every token is masked in several last frames).
>
> This is an interesting visualization suggestion. We will include such qualitative predictions in the final version.
>
> >Did you try to use masking while finetuning for a supervised task? (to obtain speed-up in computation on this stage as well). Maybe one can anneal mask ration in finetuning stage to obtain the best metrics with minimum computational resources.
>
> Applying masking during fine-tuning is a great idea. We have explored this with a masking ratio of 75% that is annealed to 0% with a cosine schedule during fine-tuning. The result is 83.8% (instead of 84.4% for full fine-tuning without masking). If we start fine-tuning with a masking ratio of 50% and anneal it to 0%, the accuracy is 84.1%. The experiments are summarized in Table A below. We think this is an encouraging result and with more tuning might be even more competitive to the original fine-tuning, but at lower cost. Thanks for this suggestion! We will include the experiment in the paper.
>
> | starting mask ratio      | K400 accuracy | speedup |
> | ----------- | ----------- | ----------- |
> | 0% (baseline)      | 84.4%       | 1.0x |
> | 50%   | 84.1%        | 1.2x |
> | 75%   | 83.8%        | 1.3x |
>
> Table A: Cosine annealing of masking ratio during fine-tuning. The starting masking ratio is varied between 0% (baseline without masking), 50% and 75%. The annealing is towards 0% at the end of fine-tuning. The model is ViT-L, with an input size of 16$\times$224$\times$224 and a spacetime patch size of 2$\times$16$\times$16. The pre-training length is 800 epochs. _Cf_. Table 2 of the main paper.
>
> > Is it possible to do Gibbs sampling with your pipeline? If yes -- did you try it and what is the quality of the samples
>
> Thanks for your suggestion. We would be happy to add experimental results if this idea could be specified further. Here are our thoughts on how we proceeded with implementing this idea: It is possible to iteratively sample and generate videos using MAE. Let $y$: MAE generated video, $x$: visible patch encodings, $m$: masked tokens. Given $x_0, m_0$, we recursively sample images with MAE's decoder by $y_{i+1}=MAE(x_i,m_i)$ and $x_{i+1}=x_i, m_{i+1}=m_{i,out}$, where $m_{i,out}$ is the output of the decoder at iteration $i$. We inspect the samples after applying MAE recursively and observe that the visual quality does not improve but rather degrades slightly after each iteration. We think the reason is that MAE is not trained for the recursive video generation approach specified above.

---

> > ### Comment · Reviewer_esSt · 2022-08-08
> > **Response to Authors**
> >
> > I thank the authors for the detailed reply and additional experiments. The results seem interesting.
> >
> > About Gibbs sampling. I was interested in whether one can sample plausible video with your model given some noise as an input. For example, complete noise in RGB space or maybe some structure noise, e.g. patches from a real video but rearranged in random order. I assume some structure should appear if one samples with Gibbs sampling with your model. I've tried it with the original MAE pipeline and discovered that there is some structure indeed but the images are not plausible. Now I believe one needs to change the training procedure to enable this Giggs-sampling behavior. So I withdraw my question.
> >
> > One more question. I wonder did you try to mix two videos together? For example, by gluing two videos and then sampling a mask that has a higher density of visible tokens at the beginning and at the end of this glued video?

---

> > > ### Author Response · Authors · 2022-08-09
> > > **Response to Reviewer esSt**
> > >
> > > Thank you for your explanation and insight into your own experiments. We also think the training procedure would need to be changed.
> > >
> > > Regarding your other question:
> > >
> > > > One more question. I wonder did you try to mix two videos together? For example, by gluing two videos and then sampling a mask that has a higher density of visible tokens at the beginning and at the end of this glued video?
> > >
> > > This is an interesting idea. We have not explored something like that but will do, thank you for suggesting it. We have noticed a possibly related work that does a similar form of mixing in the image domain https://arxiv.org/pdf/2205.13137.pdf

---

### Meta-Review · Area_Chair_1DPK · 2022-08-21

**Recommendation:** Accept
**Confidence:** Certain

**Metareview:**

This paper presents an interesting simple representation learning approach by extending masked autoencoders to videos. The reviewers have unanimously recognized the simplicity of approach, clarity of writing, and extensivity of experiments. Although there are some minor concerns about the novelty of the proposed method, the findings in this paper are of interest to the community. Given these, we are happy to recommend acceptance for this submission.

**Award:**

No

---

### Decision · Program_Chairs · 2022-09-14

Accept